# ON THE SENSITIVITY OF REWARD INFERENCE TO MISSPECIFIED HUMAN MODELS

**Joey Hong**
UC Berkeley
joey_hong@berkeley.edu

**Kush Bhatia**
Stanford *

**Anca Dragan**
UC Berkeley

## ABSTRACT

Inferring reward functions from human behavior is at the center of value alignment – aligning AI objectives with what we, humans, actually want. But doing so relies on models of how humans behave given their objectives. After decades of research in cognitive science, neuroscience, and behavioral economics, obtaining accurate human models remains an open research topic. This begs the question: *how accurate do these models need to be in order for the reward inference to be accurate?* On the one hand, if small errors in the model can lead to catastrophic error in inference, the entire framework of reward learning seems ill-fated, as we will never have perfect models of human behavior. On the other hand, if as our models improve, we can have a guarantee that reward accuracy also improves, this would show the benefit of more work on the modeling side. We study this question both theoretically and empirically. We do show that it is unfortunately possible to construct small adversarial biases in behavior that lead to arbitrarily large errors in the inferred reward. However, and arguably more importantly, we are also able to identify reasonable assumptions under which the reward inference error can be bounded linearly in the error in the human model. Finally, we verify our theoretical insights in discrete and continuous control tasks with simulated and human data.

## 1 INTRODUCTION

The expanding interest in the area of reward learning stems from the concern that it is difficult (or even impossible) to specify what we actually want AI agents to optimize, when it comes to increasingly complex, real-world tasks (Ziebart et al., 2009; Muelling et al., 2017). At the core of reward learning is the idea that human behavior serves as evidence about the underlying desired objective. Research on inferring rewards typically uses noisy-rationality as a model for human behavior: the human will take higher value actions with higher probability. It has enjoyed great success in a variety of reward inference applications (Ziebart et al., 2008; Vasquez et al., 2014; Wulfmeier et al., 2015), but researchers have also started to come up against its limitations (Reddy et al., 2018). This is not surprising, given decades of research in *behavioral* economics that has identified a deluge of systematic biases people have when making decisions on how to act, like myopia/hyperbolic discounting (Grüne-Yanoff, 2015), optimism bias (Sharot et al., 2007), prospect theory (Kahneman & Tversky, 2013), and many more (Thompson, 1999; Do et al., 2008). Hence, the noisy-rationality model has become a complication in many reward learning tasks AI researchers are interested in. For instance, in shared autonomy (Javdani et al., 2015), a human operating a robotic arm may behave suboptimally due to being unfamiliar with the control interface or the robot's dynamics, leading to the robot inferring the wrong goal (Reddy et al., 2014; Chan et al., 2021).

Recent work in reward learning attempts to go beyond noisy rationality and consider more accurate models of human behavior, by for instance looking at biases as variations on the Bellman update (Chan et al., 2021), modeling the human's false beliefs (Reddy et al., 2018), or learning their suboptimal perception process (Reddy et al., 2020). And while we might be getting closer, we will realistically never have a *perfect* model of human behavior. This raises an obvious question: *Does the human model need to be perfect in order for reward inference to be successful?* On the one hand, if small errors in the model can lead to catastrophic error in inference, the entire framework of reward learning seems ill-fated, especially as it applies to value alignment: we will never have perfect models, and we will therefore never have guarantees that the agent does not do something catastrophically bad with

---

*Work conducted as student at UC Berkeley

respect to what people actually value. On the other hand, if we can show that as our models improve, we have a guarantee that reward accuracy also improves, then there is hope: though modeling human behavior is difficult, we know that improving such models will make AI agents more aligned with us.

The main goal of this work is to study whether we can bound the error in inferred reward parameters by some function of the distance between the assumed and true human model, specifically the KL divergence between the two models. We study this question both theoretically and empirically. Our first result is a negative answer: we show that given a finite dataset of demonstrations, it is possible to hypothesize a true human model, under which the dataset is most likely to be generated, that is "close" to the assumed model, but results in arbitrarily large error in the reward we would infer via maximum likelihood estimation (MLE). However, we argue that though this negative scenario can arise, it is unlikely to occur in practice. This is because the result relies on an adversarial construction of the human model, and though the dataset is most likely, it is not necessarily representative of datasets sampled by the model. Given this, our main result is thus a reason for hope: we identify mild assumptions on the true human behavior, under which we can actually bound the error in inferred reward parameters linearly by the error of the human model. Thus, if these assumptions hold, refining the human model will monotonically improve the accuracy of the learned reward. We also show how this bound simplifies for particular biases like false internal dynamics or myopia.

Empirically, we also show a similar, optimistic message about reward learning, using both diagnostic gridworld domains (Fu et al., 2019b), as well as the Lunar Lander game, which involves continuous control over a continuous state space. First, we verify that under various simulated biases, when the conditions on the human model are likely to be satisfied, small divergences in human models do not lead to large reward errors. Second, using real human demonstration data, we derive a natural human bias and demonstrate that the same finding holds even with real humans. Overall, our results suggest an optimistic perspective on the framework of reward learning, and that efforts in improving human models will further enhance the quality of the inferred rewards.

## 2 RELATED WORK

Inverse reinforcement learning (IRL) aims to use expert demonstrations, often from a human, to infer a reward function (Ng & Russel, 2000; Ziebart et al., 2008). Maximum-entropy (MaxEnt) IRL is a popular IRL framework that models the demonstrator as noisily optimal, maximizing reward while also randomising actions as much as possible (Ziebart et al., 2008; 2010). This is equivalent to modeling humans as Boltzmann rational. MaxEnt IRL is preferred in practice over Bayesian IRL (Ramachandran & Amir, 2007), which learns a posterior over reward functions rather than a point estimate, due to better scaling in high-dimensional environments (Wulfmeier et al., 2015). More recently, Guided Cost Learning (Finn et al., 2016) and Adversarial IRL (Fu et al., 2018) learn reward functions more robust to environment changes, but build off similar modeling assumptions as MaxEnt IRL. Gleave & Toyer (2022) connected MaxEnt IRL to maximum likelihood estimation (MLE), which is the framework that we consider in this work. One of the challenges with IRL is that rewards are not always uniquely identified from expert demonstrations (Cao et al., 2021; Kim et al., 2021). Since identifiability is orthogonal to the main message of our work–sensitivity to misspecified human models–we assume that the dataset avoids this ambiguity.

Recent IRL algorithms attempt to account for possible irrationalities in the expert (Evans et al., 2016; Reddy et al., 2018; Shah et al., 2019). Reddy et al. (2018; 2020) consider when experts behave according to an internal physical and belief dynamics, and show that explicitly learning these dynamics improves accuracy of the learned reward. Singh et al. (2018) account for human risk sensitivity when learning the reward. Shah et al. (2019) propose learning general biases using demonstrations across similar tasks, but conclude that doing so without prior knowledge is difficult. Finally, Chan et al. (2021) show that knowing the type of irrationality the expert exhibits can improve reward inference over even an optimal expert. In this work, we do not assume the bias can be uncovered, but rather analyze how sensitive reward inference is to such biases.

More generally, reward learning is a specific instantiation of an inverse problem, which is well-studied in existing literature. In the framework of Bayesian inverse problems, prior work has analyzed how misspecified likelihood models affect the accuracy of the inferred quantity when performing Bayesian inference. Owhadi et al. (2015) showed that two similar models can lead to completely opposite inference of the desired quantity. Meanwhile, Sprungk (2020) showed inference is stable under a

different measure of distance between models. In this work, we also derive both instability and stability results, but consider the different problem of reward learning using MLE.

## 3 PROBLEM SETUP

**Reward parameters.** We consider Markov decision processes (MDP), which are defined by a tuple $(\mathcal{S}, \mathcal{A}, P, r, \gamma)$. Here, $\mathcal{S}, \mathcal{A}$ represent state and action spaces, $P(s'|s, a)$ and $r(s, a)$ represent the dynamics and reward function, and $\gamma \in (0, 1)$ represents the discount factor. In this work, we are interested in the setting where the reward function $r$ is unknown and needs to be inferred by a learner. We assume rewards are bounded $|r(s, a)| \leq R_{\max}$. We assume that the reward can be parameterized by *reward function parameters* $\theta \in \Theta$. We denote by $r(\cdot; \theta)$ the reward function with $\theta$ as parameters.

**True vs. assumed human policy.** Instead of having access to the reward, we observe the behavior of an "expert" demonstrator. Let $\pi^* : \Theta \times \mathcal{S} \to \Delta(\mathcal{A})$ be the reward-conditioned *demonstrator policy*, and $\mathcal{D} = \{(s_t, a_t)\}_{t=1}^n$ be a *dataset* of demonstrations provided to the learner, sampled from $\pi^*$. We use $(s, a) \sim w^\pi$ to denote observations generated by policy $\pi$, where $w^\pi$ denotes the discounted stationary distribution. We shorthand $w^{\pi^*}$ as $w^*$. Finally, let $\widetilde{\pi} : \Theta \times \mathcal{S} \to \Delta(\mathcal{A})$ be the *model* that the learner assumes generated the dataset. In practice, $\widetilde{\pi}$ is often the Boltzmann rational policy (Ziebart et al., 2008), while $\pi^*$ is an irrational policy based on human biases.

**Reward inference using the assumed policy.** Many popular algorithms in inverse reinforcement learning (IRL) (Ziebart et al., 2008; 2010) infer the reward function parameters via maximum-likelihood estimation (MLE). This is because unlike Bayesian IRL methods that learn a posterior over rewards, such MLE methods are shown to scale to high-dimensional environments (Wulfmeier et al., 2015). Using a dataset $\mathcal{D}$, the learner would estimate parameters

$$\widetilde{\theta} = \arg\min_\theta \frac{1}{n} \sum_{t=1}^n -\log \widetilde{\pi}(a_t \mid s_t; \theta) := \arg\min_\theta L(\theta; \widetilde{\pi}, \mathcal{D}). \tag{1}$$

Let $\theta^*$ be the true reward function parameters. Though $\theta^*$ cannot always be uniquely determined (Cao et al., 2021; Kim et al., 2021), for simplicity of analysis, we assume that $\theta^*$ is identifiable:

**Assumption 1.** *There exists a unique $\theta^*$ satisfying $\theta^* = \arg\min_\theta L(\theta; \pi^*, \mathcal{D})$.*

Though Assumption 1 is rather strong, we make it only because we view identifiability as orthogonal to the subject of our work – sensitivity to misspecified models.

**Goal: effect of error in the model on the error in the inferred reward.** The goal of our paper is to answer whether we can bound the distance between the inferred reward and the true reward, $d_\theta(\theta^*, \widetilde{\theta})$, as a function of the distance between the assumed human model and the true human policy, $d_\pi(\pi^*, \widetilde{\pi})$, for some useful notions of distance. If so, then we know that more accurate policies will monotonically improve the fidelity of the learned rewards. We discuss our choice of distances below.

**Reward inference error.** The inferred reward is typically used to optimize a policy in a test environment, and ultimately evaluated using the policy's performance in the new environment (Ng & Russel, 2000; Ziebart et al., 2008). However, the reason we infer reward (instead of simply cloning the human policy) is because we do not necessarily know the test environment – having the reward means we can optimize it in environments with different dynamics or initial state distributions. And unfortunately, adversarial environments exist whose dynamics amplify small disagreements between the learned and true reward functions. Hence, without prior knowledge about the test environment, we cannot derive any bound on a performance-based distance metric. We thus focus on analysis directly on a distance between the rewards themselves, via their parameters, $d_\theta(\theta^*, \widetilde{\theta}) = \|\widetilde{\theta} - \theta^*\|_2^2$. We choose squared distance as a natural and general distance metric, but admit that rewards may not be smooth in their parameters. As future work, we can improve our work using more robust measures of reward similarity (Gleave et al., 2021).

**Human model error.** Since policies are probability distributions, we can measure error in the human model as the KL-divergence between the model and demonstrator policies. We consider two different instatiations of policy divergence. The first is a *worst-case policy divergence* that takes the supremum over all reward parameters *and* states:

$$d_\pi^{\mathsf{wc}}(\pi^*, \widetilde{\pi}) = \sup_{\theta \in \Theta} \sup_{s \in \mathcal{S}} D_{\mathrm{KL}}(\pi^*(\cdot \mid s; \theta) \| \widetilde{\pi}(\cdot \mid s; \theta)). \tag{2}$$

We use the forward direction of KL-divergence because it contains an expectation over the demonstrator policy, which aligns with the sampling distribution of the dataset. Also note that this direction implies that the human model must cover actions of the true human to have small divergence. Finally, note that a small model error entails having small error across all states. This is thus a strong metric for a lower bound. It is not, however, a strong metric for an upper bound: when used in an upper bound, it would allow the model to be wrong on states and rewards without paying any more penalty than a model that is only wrong on one state and reward. We thus also consider an average error that makes for a stronger upper bound – it considers the model only on the true reward parameters $\theta^*$ and takes an *expectation over states*. We term this the *weighted policy divergence*:

$$d_\pi^{\mathsf{w}}(\pi^*, \widetilde{\pi}) = \mathbb{E}_{s \sim w^*} \left[ D_{\mathrm{KL}}(\pi^*(\cdot \mid s; \theta^*) \parallel \widetilde{\pi}(\cdot \mid s; \theta^*)) \right] . \tag{3}$$

The weighted policy divergence only looks at the states visited under the true human behavioral policy $\pi^*$ as compared to the worst case divergence, which compares against all states and rewards.

## 4    THE BAD NEWS: INSTABILITY RESULTS

We begin our theoretical analysis with a negative result. We prove that even under the worst-case policy divergence from eq. (2), a small difference between the model and the true human policy $d_\pi^{\mathsf{wc}}(\pi^*, \widetilde{\pi}) < \varepsilon$ can lead to a large inference error $d_\theta(\theta^*, \widetilde{\theta})$. Since a trivial way for this to happen is to get "unlucky" with the data set $\mathcal{D}$ drawn from $\pi^*$, and contain actions that are unlikely even if $\pi^* = \widetilde{\pi}$, we strengthen the result by excluding tail events and imposing the strong requirement that $\mathcal{D}$ contains the most likely actions under the demonstrator policy and the true reward:

**Definition 1.** *A policy $\pi$ "likely generates" $\mathcal{D}$ if for every $(s_t, a_t) \in \mathcal{D}$, the observed action is the most likely one under the true reward,* i.e., $a_t = \arg \max \pi(a_t \mid s_t; \theta^*)$.

This means that actions that appear in $\mathcal{D}$ are always at the modes of the policy $\pi^*$. Our theorem shows that despite the two policies being close on each state and reward pair, under the stronger notion of worst-case policy divergence, the inference procedure can lead to large errors.

**Theorem 2.** *For any MDP $\mathcal{M}$ with continuous actions, policy error $\varepsilon > 0$, assumed model $\widetilde{\pi}$, and dataset $\mathcal{D}$, there exists a demonstrator policy $\pi^*$ that likely generates $\mathcal{D}$ such that the worst-case policy divergence satisfies $d_\pi^{\mathsf{wc}}(\pi^*, \widetilde{\pi}) < \varepsilon$, but the reward inference error satisfies*

$$\|\widetilde{\theta} - \theta^*\|_2^2 > \frac{1}{2} \sup_{\theta, \theta' \in \Theta} \|\theta - \theta'\|_2^2 .$$

The theorem shows that for any continuous-action MDP, and any observed dataset $\mathcal{D}$, even a small perturbation in the assumed human model can lead to large inference error. We are able to prove such a strong result by perturbing the policy $\pi^*$ on only the *observed* state-action pairs in the dataset. We defer the proof of the theorem to Appendix C, and provide an illustrative example in Appendix A. Though the theorem presents a pessimistic worse-case scenario, there are aspects of the proof of it that make it impractical and potentially unlikely to occur in practice. First, the construction of $\pi^*$ is adversarial, by biasing the demonstrator at exactly the actions in the dataset. Second, though Definition 1 means the dataset is most likely to be generated by the demonstrator, it may contain actions that are not representative of what they would take. Since actions are continuous, this means that actions that are similar to the observed one (via some distance metric) are in fact very unlikely.

## 5    THE GOOD NEWS: A STABILITY RESULT

Theorem 2 paints a pessimistic picture on the feasibility of reward inference from human demonstrations, but as discussed towards the end of Section 4, the mechanisms used to derive Theorem 2 may be impractical and unlikely to occur. In this section, we show that under reasonable assumptions on the true policy we can indeed obtain a positive stability result wherein we can upper bound the reward inference error by a linear function of the weighted policy error. We identify the following assumption that enables such a result:

**Assumption 2.** *The true and model policies $\pi^*, \widetilde{\pi}$ are strongly log-concave with respect to reward parameters $\theta \in \Theta$. Formally, there exists constant $c > 0$ such that for any $s \in \mathcal{S}, a \in \mathcal{A}$, $\pi^*$ satisfies*

$$\log \pi^*(a \mid s; \theta') \leq \log \pi^*(a \mid s; \theta) + \nabla_\theta \log \pi^*(a \mid s; \theta)^\top (\theta - \theta') - \frac{c}{2} \|\theta - \theta'\|_2^2 ,$$

*and analogously for $\widetilde{\pi}$.*

The adversarial construction of demonstrator policy $\pi^*$ in deriving Theorem 2 violate the above log-concavity assumption as they involve drastic perturbations of the probabilities of just the observed actions, creating policies that are not smooth in their parameters.

**Intuition.** We know that log-concavity is violated by unnatural, adversarial constructions, but, we aim to answer: *does log-concavity always hold outside of such contrived examples?* Intuitively, we notice that log-concavity holds only if, as the reward parameter increases, an action that has become less preferred cannot become more preferred in the future. This appears to be a natural property of many policies. For example, if someone already prefers chocolate over vanilla, increasing the reward of chocolate will never cause the person to prefer vanilla over chocolate. However, it turns out there are simple problems where this is violated. In Appendix B, we present a simple navigation example where Assumption 2 is violated. Though counter examples exist, we show in our experiments that many natural biases still result in human models that satisfy Assumption 2.

Under Assumption 2, we can show that the reward inference error can be bounded linearly by weighted policy divergence. We state the formal result below, and defer its proof to Appendix C.

**Theorem 3.** *Under Assumption 2 with parameter $c > 0$, for any policies $\pi^*, \widetilde{\pi}$ with corresponding MLE reward parameters $\widetilde{\theta}, \theta^*$, the reward inference error $d_\theta(\theta^*, \widetilde{\theta})$ is bounded as*

$$\mathbb{E}_{\mathcal{D} \sim \pi^*} \left[ \|\widetilde{\theta} - \theta^*\|_2^2 \right] \leq \frac{2}{c} \mathbb{E}_{s \sim w^*} \left[ D_{KL}(\pi^*(\cdot \mid s; \theta^*) \,\|\, \widetilde{\pi}(\cdot \mid s; \theta^*)) \right] .$$

Theorem 3 differs from Theorem 2 in two important ways: (1) the reward inference error is in expectation over sampled datasets, and (2) the policy divergence is the weighted policy divergence. Both these properties are desirable, as we are agnostic to tail events due to randomness in dataset sampling, and as discussed in Section 3, we use the smaller of the two notions of divergence in the upper bound.

## 5.1 INSTANTIATING THE UPPER BOUND FOR SPECIFIC BIASES

Theorem 3 shows that the reward inference error can be bounded by the weighted policy divergence between the assumed and true policies. To understand the result in more detail, we now consider different systematic biases that could appear in human behavior, and show how they affect the weighted policy divergence and thus the upper bound.

Without loss of generality, we parameterize both the true and assumed policies as acting noisily optimal with respect to their own "Q-functions", *i.e.*, $\pi^*(a \mid s; \theta) \propto \exp(Q^*(s, a; \theta))$ and $\widetilde{\pi}(a \mid s; \theta) \propto \exp(\widetilde{Q}(s, a; \theta))$. Importantly, note that even though this parameterization is used in MaxEnt IRL with the soft Q-values (Ziebart et al., 2008; 2010), neither $Q^*$ nor $\widetilde{Q}$ need necessarily be optimal – in this analysis, we will use $\widetilde{Q}$, the human model, as the soft Q-value function, and show what happens when the true model coming from $Q^*$ suffers from certain biases. Following prior work (Reddy et al., 2018; Chan et al., 2021), we examine biases that can be modelled as deviations from the Bellman update. For a tabular MDP $M$ with $|\mathcal{S}|, |\mathcal{A}| < \infty$, the soft Bellman update satisfies:

$$Q(s, a; \theta) := r(s, a; \theta) + \gamma \sum_{s'} P(s' \mid s, a) V(s; \theta), \ V(s; \theta) := \log \left( \sum_{a \in \mathcal{A}} \exp(Q(s, a; \theta)) \right) . \quad (4)$$

Formally, we study examples under which the human demonstrator's Q-values $Q^*(s, a; \theta)$ satisfy (4) but under a biased MDP $M^*$. We consider two specific sources of bias in the MDP: (1) the transition model $P$ and (2) the discounting factor $\gamma$. By parameterizing the biases in this way, we now have an intuitive notion of the degree of bias, and can study how the magnitude of the bias affects the policy divergence in (3). For brevity, we simply state the results as corollaries and defer proofs to Appendix C.

**Internal dynamics.** We first consider irrationalities that result from human demonstrators having an *internal dynamics model* $P^*$ that is misspecified. For example, studies in cognitive science have shown that humans tend to underestimate the effects of inertia in projectile motion (Caramazza et al., 1981). Similar studies have also shown that humans overestimate their control over randomness in the environment (Thompson, 1999), dubbed *illusion of control*. The latter irrationality can be formalized in our parameterization by assuming that $P^*(\cdot \mid s, a) \propto (P(\cdot \mid s, a))^n$, where as $n \to \infty$, the human

will believe the dynamics of the MDP are increasingly more deterministic. In Corollary 4, we show that the policy distance can be bounded linearly by the bias in transition dynamics:

**Corollary 4.** *Let $\Delta_P = \sup_{s,a} \| P^*(\cdot \mid s, a) - \widetilde{P}(\cdot \mid s, a)\|_1$. Also, let $\pi^*, \widetilde{\pi}$ be the policies that result from from value iteration using* (4) *with dynamics models $P^*, \widetilde{P}$, respectively. Then, their weighted policy divergence is bounded as*

$$\mathbb{E}_{s \sim d^*}\left[D_{KL}(\pi^*(\cdot \mid s; \theta^*) \| \widetilde{\pi}(\cdot \mid s; \theta^*))\right] \leq \frac{2|\mathcal{A}|R_{\max}}{(1-\gamma)^2} \Delta_P.$$

**Myopia Bias.** The other irrationality we study is when humans overvalue near-term rewards, dubbed *myopia* (Grüne-Yanoff, 2015). Such bias can be captured in our parameterization through a biased discount factor $\gamma^*$, where as $\gamma^* \to 0$, the human will act more greedily and prioritize immediate reward. In Corollary 5, we bound the distance between policies by the absolute difference in their internal discount factor.

**Corollary 5.** *Let $\pi^*, \widetilde{\pi}$ be the policies that result from value iteration using* (4) *with discount factors $\gamma^*, \widetilde{\gamma}$, respectively. Then, their weighted policy divergence is bounded as*

$$\mathbb{E}_{s \sim d^*}\left[D_{KL}(\pi^*(\cdot \mid s; \theta^*) \| \widetilde{\pi}(\cdot \mid s; \theta^*))\right] \leq \frac{2|\mathcal{A}|R_{\max}}{(1-\widetilde{\gamma})(1-\gamma^*)} |\widetilde{\gamma} - \gamma^*|.$$

The above result shows that the degree of bias linearly upper-bounds the weighted policy divergence and hence, from Theorem 3, the expected reward inference error.

## 6 EMPIRICAL ANALYSIS

Our theoretical results predict that in the worst-case, reward inference error can be arbitrarily bad relative to the human model error, but on average, under some assumptions, the reward error should be small. Our empirical results aim to complement our theory by answering the following question under natural biases in human models in various environments: *do we find a stable relationship between policy divergence and reward error?*

We tackle this in three ways: (1) simulating the specific biases we analyzed in Section 5.1, (2) simulating a non-Bellman-update structured kind of bias (a demonstrator that is still learning about the environment), and (3) collecting *real human data*. We consider both tabular navigation tasks on gridworld, as well as more challenging continuous control tasks on the Lunar Lander game (Brockman et al., 2016b).

**Experiment design.** Each experiment has a bias we study and an environment (gridworld or LunarLander). When considering simulated biases, we manipulate $\pi^*$ by manipulating the *magnitude of the bias* starting at $\pi^* = \widetilde{\pi}$ the Boltzmann optimal policy. This helps us simulate different hypothetical humans, and see what degree of deviation from optimality ends up negatively impacting reward inference. When modeling bias with real human data, we instead fix $\pi^*$ as the real human policy, and manipulate $\widetilde{\pi}$ by interpolating between the Boltzmann optimal policy and the real human policy – this emulates a practical process where human models get increasingly more accurate.

### 6.1 TABULAR EXPERIMENTS WITH STRUCTURED BIASES

First, we consider tabular navigation in gridworld domains (Fu et al., 2019a), where the task is the reach the goal state and earn a reward of $\theta > 1$, which is not known to the agent, while avoiding getting trapped at lava states. To further complicate the task, the agent can also get stuck at "waypoint" states that yield a reward of 1. Depending on the environment, it can be better for the agent to stop at the waypoint state, to circumvent taking the longer, more treacherous path to the goal state. The agent is able to move in either of the four directions, or choose to stay still. To introduce stochasticity in the transition dynamics, there is a 30% chance that the agent travels in a different direction

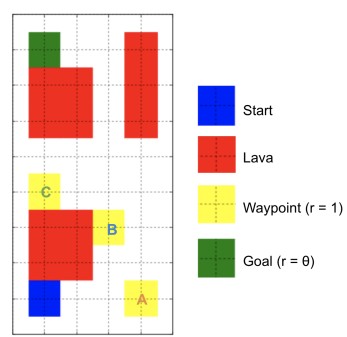

**Figure 3:** Gridworld environments.

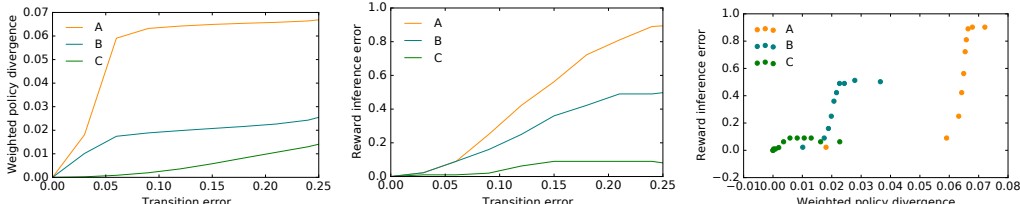

**Figure 1:** Effect of transition error (measured as the degree of underestimation of unintended transitions) on (a) weighted policy divergence and (b) reward inference error on three Gridworld environments (A,B,C). In (c), we show a scatter plot of the policy and reward errors for different biased transition model. Note that small policy divergence results in small reward inference error.

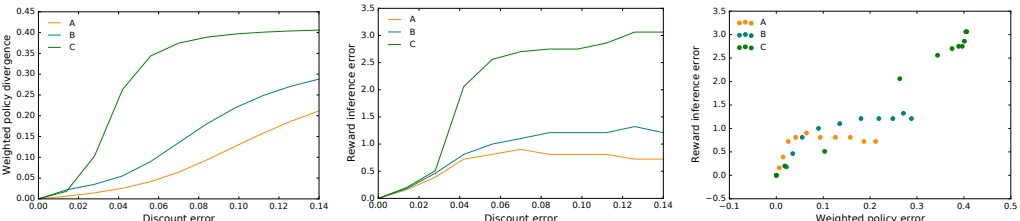

**Figure 2:** Effect of discount error in the three Gridworld environments. Like Figure 1, we see a strong correlation between policy divergence and reward inference error.

than commanded. We consider three different gridworlds (which we simply call environments A, B, and C) where we vary in the location of the waypoint state (shown in Figure 3).

In each environment, we want to the learn the underlying reward parameter $\theta$ from demonstrations; however, the model $\widetilde{\pi}$ is noisily optimal, whereas the demonstrator policy $\pi^*$ is irrational by suffering from false internal dynamics, or myopia. We model these irrationalities by either modifying the transition matrix or discount factor, respectively, in the soft Bellman update in Equation (4). Note that the stationary distribution $w^\pi$ for a policy $\pi$ can be exactly computed; hence, instead of sampling data from $\pi^*$, we use $w^*$ to compute exact quantities (see Appendix D.1 for technical details).

**Internal dynamics.** The first irrationality we consider is illusion of control, where the demonstrator policy significantly underestimates the stochasticity in the environment. Such biased policies $\pi^*$ are obtained via value iteration on a biased transition matrix $P^*$, where the human wrongly believes the probability $p$ of unintended transitions is smaller than the true value. As $p \to 0$, the demonstrator becomes more confident that they can reach the goal state, and will prefer reaching the goal over the waypoint state, even when the latter is much closer and safely reachable (see Appendix D.1 for visualizations of the biased policies). In Figure 1, we show the effect of the transition bias (error in $p$) on both the weighted policy divergence, and the reward inference error. The sub-linear trend in Figure 1a agrees with Corollary 4. Figure 1b and c show a sub-linear dependence of the reward inference error on the policy divergence, as predicted by Theorem 3. For environment A, the reward error goes up most quickly with the dynamics error, but so does the weighted policy divergence, making this divergence a better indicator of reward error than simply the dynamics error.

**Myopia.** The next irrationality we look at is myopia, where the demonstrator policy assumes a biased discounting factor $\gamma^*$ that underestimates the true one. As $\gamma^* \to 0$, the biased agent will much more strongly prefer the closer waypoint state over the goal state. In Figure 2, we see analogous results to the internal dynamics bias. Namely, Figure 2a agrees with Corollary 5, and Figure 2b and c shows a sub-linear correlation between policy and reward error, as predicted by Theorem 3.

## 6.2 CONTINUOUS CONTROL EXPERIMENTS

Next, we consider a more challenging domain of navigation with continuous states and actions. The exact navigation environment is a modification the Lunar Lander game with continuous actions (Brockman et al., 2016a), where the agent receives a reward for landing safely on the landing pad. The agent is able to take a continuous action in $[-1, 1]^2$ that encodes the directions it wants to move (left, right, up) via its sign, as well as how much power it wants to use in each direction via its cardinality.

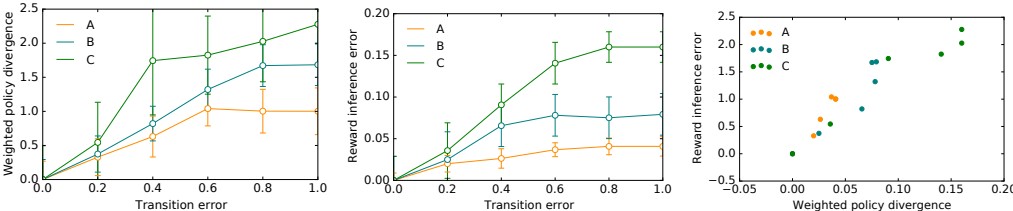

**Figure 4:** Effect of transition error (measured as error in $p$) in the continuous Lunar Lander environments. The results are consistent with earlier Gridworld results.

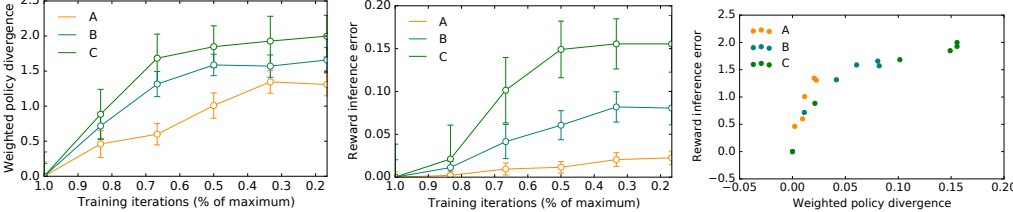

**Figure 5:** Effect of the simulated human learning bias in the continuous Lunar Lander environments.

In constrast to the classic version of the game where the landing pad is always in the middle, we vary its location. The unknown reward parameter $\theta \in (0, 1)$ is the location of the landing pad (as a horizontal displacement normalized by the total width of the environment). We consider three different environments that differ in the location of the landing pad (see Figure 6). In each environment, the human model $\widetilde{\pi}$ is the near-optimal one obtained by soft actor-critic (Haarnoja et al., 2018). We provide details on the training procedure in Appendix D.2. In these ex-periments, we simulate the internal dynamics bias as well as a new one based on the notion of a demonstrator that is still themselves learning.

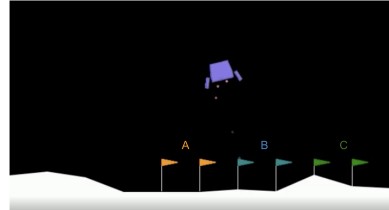

**Figure 6:** Lunar Lander environments.

**Internal dynamics.** We first study of the effect of demonstrator policies with biased dynamics models. We bias the dynamics model by varying a parameter $p$ that describes how much one unit of power will increase acceleration in the corresponding direction. This is a plausibly natural human bias as people tend to underestimate the effects of inertia in projectile motion (Caramazza et al., 1981). For each false setting of $p$, we learn a biased policy that is near-optimal for that $p$. As $p$ increase, the biased policy tends to underestimate the amount of power required to move the lander enough to the right to reach the landing pad (see Appendix D.2 for visualizations). In Figure 4, we show the effect of the transition bias (error in $p$) on both the weighted policy divergence and the reward inference error. We see that even in a challenging continuous control domain, Theorem 3 still holds.

**Demonstrators that are learning.** We next simulate a bias that might arise from humans that are learning how to do the task (as would be the case, for instance, in our Lunar Lander task). We do so by varying the amount of training iterations in learning the policy. The degree of such bias is captured in a parameter $\rho \in [0, 1]$, that denotes the number of training iterations, normalized by the amount used to learn the near-optimal believed policy. In Figure 5, we show the effect of $\rho$ on both the weighted policy divergence and the reward inference error. Reassuringly, we again notice a sub-linear correlation in line with Theorem 3.

## 6.3 ANALYSIS OF REAL HUMAN POLICIES

The previous experiments have considered natural but sim-ulated biases to construct demonstrator policies modeling biased humans. However, it remains to be seen whether Theorem 3 remains predictive even when biases are from real humans. We consider the same Lunar Lander game in Section 6.2 but discretize the action space so that people can intuitively interact in it. Using this environment, we

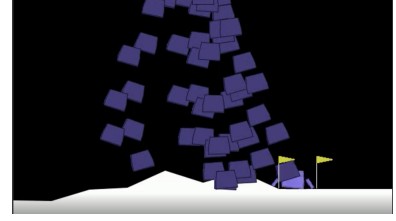

**Figure 8:** Visualization of trajectories under the human policy.

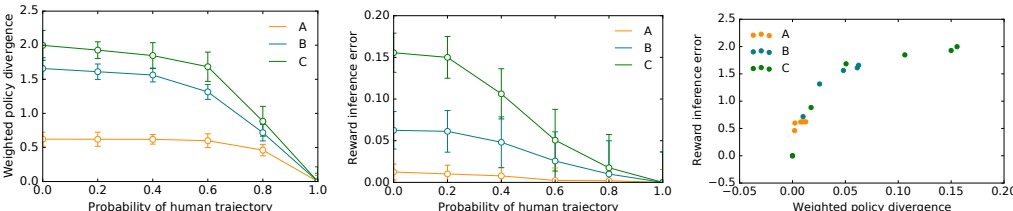

**Figure 7:** Effect of modeling human bias (measured by probability of acting according to human policy) on (a) weighted policy divergence and (b) reward inference error on discrete Lunar Lander environments. In (c), we show a scatter plot of the policy and reward errors for different probabilities. We see that more accurate human models correspond to lower reward inference error.

create a demonstrator policy grounded in real human demonstrations. We do this by collecting trajectories from 10 human demonstrators, then learning a policy that imitates human behavior by running behavior cloning (BC) on the aggregated trajectories. We visualize trajectories from this policy in Figure 8. We observe that in general, humans tend to be unable to properly account for the effect of gravity, causing them to crash the lander before it has moved enough horizontally, particularly in environments B and C where the landing pad is horizontally displaced from the middle.

Then, we emulate a process through which the learner's model of the human, $\widetilde{\pi}$, would evolve to align more and more with the true human policy. Specifically, we vary $\widetilde{\pi}$ to interpolate between near-optimal and the true human policy, while keeping $\pi^*$ fixed to the latter. To do so, we vary a parameter $\alpha$ that controls the probability of sampling from the human policy (vs. the near-optimal one). In Figure 7, we show the effect of $\alpha$ on the weighted policy divergence and reward inference error, and conclude that larger $\alpha$ result in smaller policy divergence as well as reward error. Importantly, this suggest that as the model gets closer and closer to the true human policy, the reward estimate gets increasingly better, with no sign that a small model error leads to a terrible reward. In addition, we also match the simulation experiments by keeping $\widetilde{\pi}$ fixed as the optimal policy, and interpolate between the optimal policy and the real human policy for $\pi^*$ (see Appendix D.3). This gives us hope that even in real-world problems, better human models $\widetilde{\pi}$ can translate to better reward inference.

## 7 DISCUSSION

**Summary.** In this paper, we conduct a theoretical and empirical study of how sensitive reward learning from human demonstrations is to misspecification of the human model. First, we provide an ominous result that arbitrarily small divergences in the assumed human model can result in large reward inference error. While this is in theory possible, it requires a rather adversarial construction that makes it unlikely to occur in practice. In light of this, we identify assumptions under which the reward error can actually be upper-bounded linearly by the model error. Experiments with multiple biases in different environments, as well as an analysis of the true human policy (which potentially suffers from unknown, yet to be characterized suboptimalities), reassuringly show remarkably consistent results: over and over again, we see that as the human model and the true human behavior are more and more aligned, the reward error decreases. Overall, our results convey the optimistic message that reward learning improves as we obtain better human models, and motivate further research into improved models.

**Limitations and future work.** Our upper-bound relies on Assumption 2 of log-concavity of the human policy. However, we hypothesize that weaker assumptions exist from which we can derive similar bounds as Theorem 3. In addition, via Assumption 1, we ignore ambiguity in reward identification. It may be important in the future to consider reward error and identifiability jointly, potentially through equivalence classes of reward functions. Finally, as alluded to in Section 3, more robust measures of reward similarity exist that avoid needing to implicitly assume smoothness of the reward function in their parameters (Gleave et al., 2021). Interesting directions of further investigation include: (1) how to better model human biases, or orthogonally, (2) how to modify existing reward inference algorithms to be more robust to misspecification. A negative side-effect of our work could occur when we mistakenly rely on the upper-bound in Theorem 3 when its conditions are not met, *i.e.*, Assumption 2 does not hold, resulting in catastrophically bad inference without knowing it. More broadly, reward learning in general has the issue that it does not specify *whose* reward to learn, and how to combine different people's values.

## ACKNOWLEDGEMENTS

We thank the members of the InterACT lab, and BAIR as a whole, at UC Berkeley for their helpful discussions, as well as providing human demonstrations for our experiments. We also thank anonymous reviewers for feedback on an early version of this paper. This research is funded in part by the Office of Naval Research Young Investigator Program, and UCSF Weill Institute for Neurosciences. Additionally, KB was supported in part by a grant from Long-Term Future Fund (LTFF).

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

## A    EXAMPLE WHERE LOWER-BOUND OCCURS

Let us consider a stochastic bandit with continuous actions $\mathcal{A} \in [0,1]$. Since bandits consist of a single, stationary state, we drop dependence on state in all quantities. The reward for choosing action $a \in \mathcal{A}$ is $r(a;\theta) = a^\theta (1-a)^{1-\theta}$, for some parameter $\theta \in (0,1)$. When $\theta$ is close to 0, the reward is higher for actions close to 0, and vice-versa when $\theta$ is close to 1.

For simplicity, let us only consider a dataset of a single action $a_1$. Let us consider a Boltzmann rational policy as the assumed model, namely $\widetilde{\pi}(a;\theta) \propto \exp(r(a;\theta))$ and have the demonstrator policy $\pi^*$ be an adversarial perturbation of $\widetilde{\pi}$ that overestimates the reward of $a_1$:

$$\pi^*(a;\theta) \propto \begin{cases} \exp(r(a;\theta)) \left( \mathbb{1}\{a \notin (a_1 - \frac{\delta}{2}, a_1 + \frac{\delta}{2})\} + 10^9 \, \mathbb{1}\{a \in (a_1 - \frac{\delta}{2}, a_1 + \frac{\delta}{2})\} \right) & \text{if } \theta < 0.001 \\ \exp(r(a;\theta)) & \text{otherwise}, \end{cases}$$

for some $\delta \in (0,1)$. The interpretation of this is that the human is believed to be noisily optimal; however, the human actually overestimates the value of an infinitesimal region centered at action $a_1$ only if $\theta$ is close to 0. Note that $d_\pi^{\mathsf{wc}}(\pi^*, \widetilde{\pi}) < c\delta$ for some constant $c$, so we can choose $\delta$ such that the two policies are "close" to each other. When $a_1 = 1$, we will infer $\widetilde{\theta} \approx 1$; however, $\theta^* \approx 0$, leading to reward inference error equal to the range of reward parameters.

## B    EXAMPLE WHERE LOG-CONCAVITY IS VIOLATED

The environment is a $3 \times 3$ gridworld with deterministic transitions and discount $\gamma = 1$. Let $s$ be the center cell, and $a$ be going up. In Figure 9, we show that a natural policy that chooses "up" according to $\pi(a \mid s;\theta) \propto \exp(\max(\theta, 10 - \theta))$ violates log-concavity. The reason is that "up" is optimal for $\theta \in [0,4] \cup [6,10]$ but not in between.

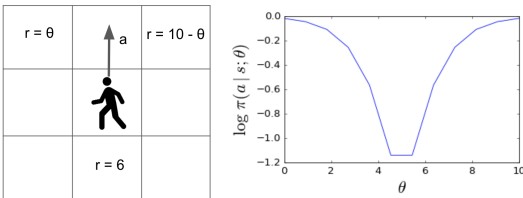

**Figure 9:** Simple navigation environment where a near optimal policy violates Assumption 2.

## C    PROOFS

### C.1    PROOF OF THEOREM 2

Without loss of generality, let $\widetilde{\pi}$ satisfy $\widetilde{\pi}(a \mid s;\theta) = \exp(\Phi(s,a;\theta))/Z(s;\theta)$. Note that this parameterization can be used to express any probability distribution. Also, for any $\delta > 0$, let us define

$$B_\delta(\mathcal{D}) = \bigcup_{t=1}^n \left( a_t - \frac{\delta}{2}, a_t + \frac{\delta}{2} \right)$$

as a union of $\frac{\delta}{2}$-balls around the actions that appear in dataset $\mathcal{D}$. Then, for any $\theta^* \in \Theta$, let $\pi^*$ satisfy

$$\pi^*(a \mid s;\theta) = \begin{cases} \frac{1}{Z'(s;\theta)} \left( \exp(\Phi(s,a;\theta)) \mathbb{1}\{a \notin B_\delta(\mathcal{D})\} + C \, \mathbb{1}\{a \in B_\delta(\mathcal{D})\} \right) & \text{if } \theta = \theta^*, \\ \frac{1}{Z(s;\theta)} \exp(\Phi(s,a;\theta)) & \text{otherwise}, \end{cases}$$

where $C$ is the supremum $C = \sup_{s,a,\theta} \exp(\Phi(s,a;\theta))$. By construction, it is clear that MLE on $\mathcal{D}$ using $\pi^*$ would yield reward parameter $\theta^*$. Since such $\pi^*$ can be constructed for any $\theta^*$, we can choose $\theta^*$ that satisfies $\|\widetilde{\theta} - \theta^*\|_2^2 > \sup_{\theta,\theta'} \|\theta - \theta'\|_2/2$. What remains is showing that there exists $\delta$ such that $d_\pi^{\mathsf{wc}}(\pi^*, \widetilde{\pi}) < \varepsilon$.

**Bounding the worst-case policy divergence.** Note that the worst-case divergence is necessarily satisfied at $\theta^*$. Fix any state $s \in \mathcal{S}$. We have,

$$D_{\text{KL}}(\pi^*(\cdot \mid s; \theta^*) \| \widetilde{\pi}(\cdot \mid s; \theta^*)) = \int_{\mathcal{A}} \pi^*(a \mid s; \theta^*) \log \frac{\pi^*(a \mid s; \theta^*)}{\widetilde{\pi}(a \mid s; \theta^*)} da$$

$$= \int_{\mathcal{A} \setminus B_\delta(\mathcal{D})} \pi^*(a \mid s; \theta^*) \log \frac{\pi^*(a \mid s; \theta^*)}{\widetilde{\pi}(a \mid s; \theta^*)} da + \int_{B_\delta(\mathcal{D})} \pi^*(a \mid s; \theta^*) \log \frac{\pi^*(a \mid s; \theta^*)}{\widetilde{\pi}(a \mid s; \theta^*)} da .$$

We consider each term individually. Starting with the first term, we have

$$\int_{\mathcal{A} \setminus B_\delta(\mathcal{D})} \pi^*(a \mid s; \theta^*) \log \frac{\pi^*(a \mid s; \theta^*)}{\widetilde{\pi}(a \mid s; \theta^*)} da \leq \log \frac{Z(s; \theta^*)}{Z'(s; \theta^*)}$$

$$\leq \frac{1}{Z(s; \theta^*)} |Z(s; \theta^*) - Z'(s; \theta^*)| ,$$

where we use that the policies only differ in their normalizers in the first inequality, and that $\log(t) - \log(s) \leq \frac{1}{\min\{t, s\}} |t - s|$ in the second. Now, using that

$$Z'(s; \theta^*) = \int_{\mathcal{A} \setminus B_\delta(\mathcal{D})} \exp(\Phi(s, a, ; \theta^*)) da + C |B_\delta(\mathcal{D})| ,$$

we have

$$\int_{\mathcal{A} \setminus B_\delta(\mathcal{D})} \pi^*(a \mid s; \theta^*) \log \frac{\pi^*(a \mid s; \theta^*)}{\widetilde{\pi}(a \mid s; \theta^*)} da \leq \frac{C}{Z(s; \theta^*)} |B_\delta(\mathcal{D})| .$$

Now, let us consider the second term. we have

$$\int_{B_\delta(\mathcal{D})} \pi^*(a \mid s; \theta^*) \log \frac{\pi^*(a \mid s; \theta^*)}{\widetilde{\pi}(a \mid s; \theta^*)} da \leq \int_{B_\delta(\mathcal{D})} \pi^*(a \mid s; \theta^*) |\log C - \Phi(s, a; \theta^*)| da + \log \frac{Z(s; \theta^*)}{Z'(s; \theta^*)}$$

$$\leq 2 \log C |B_\delta(\mathcal{D})| + \frac{C}{Z(s; \theta^*)} |B_\delta(\mathcal{D})| ,$$

where we reuse the bound for the first term, and use that $|\phi(s, a; \theta^*)| \leq \log C$. Combining the two bounds yields

$$D_{\text{KL}}(\pi^*(\cdot \mid s; \theta^*) \| \widetilde{\pi}(\cdot \mid s; \theta^*)) \leq 2 \log C |B_\delta(\mathcal{D})| + \frac{2C}{Z(s; \theta^*)} |B_\delta(\mathcal{D})| .$$

Using that $|B_\delta(\mathcal{D})| \leq n\delta$ by construction, we can solve for $\delta = \mathcal{O}(\varepsilon/n)$ such that $d_\pi^{\text{wc}}(\pi^*, \widetilde{\pi}) < \varepsilon$, as desired. This completes the proof.

### C.2 PROOF OF THEOREM 3

Recall that $L(\theta; \pi, \mathcal{D})$ is the negative log-likelihood of demonstrations $\mathcal{D}$ under policy $\pi$ and reward parameters $\theta$. Note that we can write

$$L(\theta; \widetilde{\pi}, \mathcal{D}) = \frac{1}{n} \sum_{t=1}^{n} -\log \widetilde{\pi}(a_t \mid s_t; \theta) = \frac{1}{n} \sum_{t=1}^{n} -\log \pi^*(a_t \mid s_t; \theta) + \log \frac{\pi^*(a_t \mid s_t; \theta)}{\widetilde{\pi}(a_t \mid s_t; \theta)}$$

$$= L(\theta; \pi^*, \mathcal{D}) + \frac{1}{n} \sum_{t=1}^{n} \log \frac{\pi^*(a_t \mid s_t; \theta)}{\widetilde{\pi}(a_t \mid s_t; \theta)} .$$

By the law of large numbers, we have that under expectation over dataset $\mathcal{D}$,

$$\mathbb{E}_{\mathcal{D} \sim \pi^*} \left[ \frac{1}{n} \sum_{t=1}^{n} \log \frac{\pi^*(a_t \mid s_t; \theta)}{\widetilde{\pi}(a_t \mid s_t; \theta)} \right] = \mathbb{E}_{s \sim d^*} \left[ D_{\text{KL}}(\pi^*(\cdot \mid s; \theta^*) \| \widetilde{\pi}(\cdot \mid s; \theta^*)) \right]$$

Using Assumption 2 on $\pi^*$, for any $\theta \in \Theta$, we also have

$$L(\theta; \pi^*, \mathcal{D}) \geq L(\theta^*; \pi^*, \mathcal{D}) + \nabla_\theta L(\theta^*; \pi^*, \mathcal{D})^\top (\theta^* - \theta) + \frac{cn}{2} \|\theta - \theta^*\|_2^2 .$$

By definition of $\theta^*$ and Assumption 1, we know that $\nabla_\theta L(\theta^*; \pi^*, \mathcal{D}) = \mathbf{0}$. Substituting $\theta = \widetilde{\theta}$ and rearranging yields

$$\|\widetilde{\theta} - \theta^*\|_2^2 \leq \frac{2}{c} \left( L(\widetilde{\theta}; \pi^*, \mathcal{D}) - L(\theta^*; \pi^*, \mathcal{D}) \right) .$$

Analogously, using Assumption 2 on $\widetilde{\pi}$ [1], we have that

$$\|\widetilde{\theta} - \theta^*\|_2^2 \leq \frac{2}{c} \left( L(\theta^*; \widetilde{\pi}, \mathcal{D}) - L(\widetilde{\theta}; \widetilde{\pi}, \mathcal{D}) \right) .$$

Combining the two bounds yields,

$$\|\widetilde{\theta} - \theta^*\|_2^2 \leq \frac{2}{c} \left( L(\theta^*; \widetilde{\pi}, \mathcal{D}) - L(\theta^*; \pi^*, \mathcal{D}) + L(\widetilde{\theta}; \pi^*, \mathcal{D}) - L(\widetilde{\theta}; \widetilde{\pi}, \mathcal{D}) \right)$$

$$\leq \frac{2}{c} \left( \frac{1}{n} \sum_{t=1}^n \log \frac{\pi^*(a_t \mid s_t; \theta^*)}{\widetilde{\pi}(a_t \mid s_t; \theta^*)} - \frac{1}{n} \sum_{t=1}^n \log \frac{\pi^*(a_t \mid s_t; \widetilde{\theta})}{\widetilde{\pi}(a_t \mid s_t; \widetilde{\theta})} \right)$$

Taking an expectation over dataset $\mathcal{D}$ yields the desired result

$$\mathbb{E}\left[\|\widetilde{\theta} - \theta^*\|_2^2\right] \leq \frac{2}{c}\mathbb{E}_{s \sim d^*} \left[ D_{\mathrm{KL}}(\pi^*(\cdot \mid s; \theta^*) \,\|\, \widetilde{\pi}(\cdot \mid s; \theta^*)) - D_{\mathrm{KL}}(\pi^*(\cdot \mid s; \widetilde{\theta}) \,\|\, \widetilde{\pi}(\cdot \mid s; \widetilde{\theta})) \right]$$

$$\leq \frac{2}{c}\mathbb{E}_{s \sim d^*} \left[ D_{\mathrm{KL}}(\pi^*(\cdot \mid s; \theta^*) \,\|\, \widetilde{\pi}(\cdot \mid s; \theta^*)) \right] ,$$

which is the desired result.

## C.3 PROOF OF COROLLARY 4

Recall that $\widetilde{\pi}, \pi^*$ are parameterized by Q-values $\widetilde{Q}, Q^*$ that satisfy the soft Bellman update in (4). Fix state $s$. We have

$$D_{\mathrm{KL}}(\pi^*(\cdot \mid s; \theta^*)\|\widetilde{\pi}(\cdot \mid s; \theta^*)) = \mathbb{E}_{a \sim \pi^*(\cdot \mid s; \theta^*)} \left[ Q^*(s, a; \theta^*) - \widetilde{Q}(s, a; \theta^*) \right] + \log \frac{\sum_{a'} \exp(\widetilde{Q}(s, a'; \theta^*))}{\sum_{a'} \exp(Q^*(s, a'; \theta^*))} .$$

For any action $a$, we have

$$Q^*(s, a; \theta^*) - \widetilde{Q}(s, a; \theta^*) = \gamma \sum_{s'} P^*(s' \mid s, a) V^*(s'; \theta^*) - \gamma \sum_{s'} \widetilde{P}(s' \mid s, a) \widetilde{V}(s'; \theta^*)$$

$$= \gamma \sum_{s'} P^*(s' \mid s, a) V^*(s'; \theta^*) + \gamma \sum_{s'} P^*(s' \mid s, a) \widetilde{V}(s'; \theta^*)$$

$$- \gamma \sum_{s'} P^*(s' \mid s, a) \widetilde{V}(s'; \theta^*) - \gamma \sum_{s'} \widetilde{P}(s' \mid s, a) \widetilde{V}(s'; \theta^*)$$

$$= \gamma \|P^*(\cdot \mid s, a) - \widetilde{P}(\cdot \mid s, a)\|_1 \widetilde{V}(s'; \theta^*) + \gamma \sum_{s'} P^*(s' \mid s, a)(V^*(s'; \theta^*) - \widetilde{V}(s'; \theta^*))$$

$$\leq \frac{R_{\max}}{1 - \gamma} \Delta_P + \gamma \sum_{s'} P^*(s' \mid s, a) \max_{a'}\{Q^*(s', a'; \theta^*) - \widetilde{Q}(s', a'; \theta^*)\}$$

$$\leq \cdots$$

$$\leq \frac{R_{\max}}{(1 - \gamma)^2} \Delta_P .$$

Now, let us consider the normalization term. We have

$$\log \frac{\sum_{a'} \exp(\widetilde{Q}(s, a'; \theta^*))}{\sum_{a'} \exp(Q^*(s, a'; \theta^*))} \leq \frac{1}{\sum_{a'} \exp(\widetilde{Q}(s, a'; \theta^*))} \sum_{a'} \exp(\widetilde{Q}(s, a'; \theta^*)) \log \frac{\exp(\widetilde{Q}(s, a'; \theta^*))}{\exp(Q^*(s, a'; \theta^*))}$$

$$\leq \sum_{a'} (\widetilde{Q}(s, a'; \theta^*) - Q^*(s, a'; \theta^*))$$

$$\leq \frac{|\mathcal{A}| R_{\max}}{(1 - \gamma)^2} \Delta_P .$$

Combining the two bounds and taking an expectation over $s$ yields the desired result.

---

[1]In the statement of Assumption 2 in the main paper, we only assume log-concavity for $\pi^*$. This will be corrected in a future revision to include both $\pi^*, \widetilde{\pi}$

## C.4 PROOF OF COROLLARY 5

The proof follows the format of the proof for Corollary 4. Fix state $s$. We have

$$D_{\mathrm{KL}}(\pi^*(\cdot \mid s; \theta^*) \| \widetilde{\pi}(\cdot \mid s; \theta^*)) = \mathbb{E}_{a \sim \pi^*(\cdot|s;\theta^*)} \left[ Q^*(s,a;\theta^*) - \widetilde{Q}(s,a;\theta^*) \right] + \log \frac{\sum_{a'} \exp(\widetilde{Q}(s,a';\theta^*))}{\sum_{a'} \exp(Q^*(s,a';\theta^*))} .$$

For any action $a$, we have

$$
\begin{aligned}
Q^*(s,a;\theta^*) - \widetilde{Q}(s,a;\theta^*) &= \gamma^* \sum_{s'} P(s' \mid s,a) V^*(s';\theta^*) - \widetilde{\gamma} \sum_{s'} P(s' \mid s,a) \widetilde{V}(s';\theta^*) \\
&= \gamma^* \sum_{s'} P(s' \mid s,a) V^*(s';\theta^*) + \gamma^* \sum_{s'} P(s' \mid s,a) \widetilde{V}(s';\theta^*) \\
&\quad - \gamma^* \sum_{s'} P(s' \mid s,a) \widetilde{V}(s';\theta^*) - \widetilde{\gamma} \sum_{s'} P(s' \mid s,a) \widetilde{V}(s';\theta^*) \\
&= (\gamma^* - \widetilde{\gamma}) \sum_{s'} P(s' \mid s,a) \widetilde{V}(s';\theta^*) + \gamma^* \sum_{s'} P(s' \mid s,a)(V^*(s';\theta^*) - \widetilde{V}(s';\theta^*)) \\
&\leq \frac{R_{\max}}{1 - \widetilde{\gamma}} |\gamma^* - \widetilde{\gamma}| + \gamma^* \sum_{s'} P(s' \mid s,a) \max_{a'}\{Q^*(s',a';\theta^*) - \widetilde{Q}(s',a';\theta^*)\} \\
&\leq \dots \\
&\leq \frac{R_{\max}}{(1 - \widetilde{\gamma})(1 - \gamma^*)} |\gamma^* - \widetilde{\gamma}| .
\end{aligned}
$$

Now, let us consider the normalization term. We have

$$
\begin{aligned}
\log \frac{\sum_{a'} \exp(\widetilde{Q}(s,a';\theta^*))}{\sum_{a'} \exp(Q^*(s,a';\theta^*))} &\leq \frac{1}{\sum_{a'} \exp(\widetilde{Q}(s,a';\theta^*))} \sum_{a'} \exp(\widetilde{Q}(s,a';\theta^*)) \log \frac{\exp(\widetilde{Q}(s,a';\theta^*))}{\exp(Q^*(s,a';\theta^*))} \\
&\leq \sum_{a'} (\widetilde{Q}(s,a';\theta^*) - Q^*(s,a';\theta^*)) \\
&\leq \frac{|\mathcal{A}| R_{\max}}{(1 - \widetilde{\gamma})(1 - \gamma)} |\widetilde{\gamma} - \gamma^*| .
\end{aligned}
$$

Combining the two bounds yields the desired result.

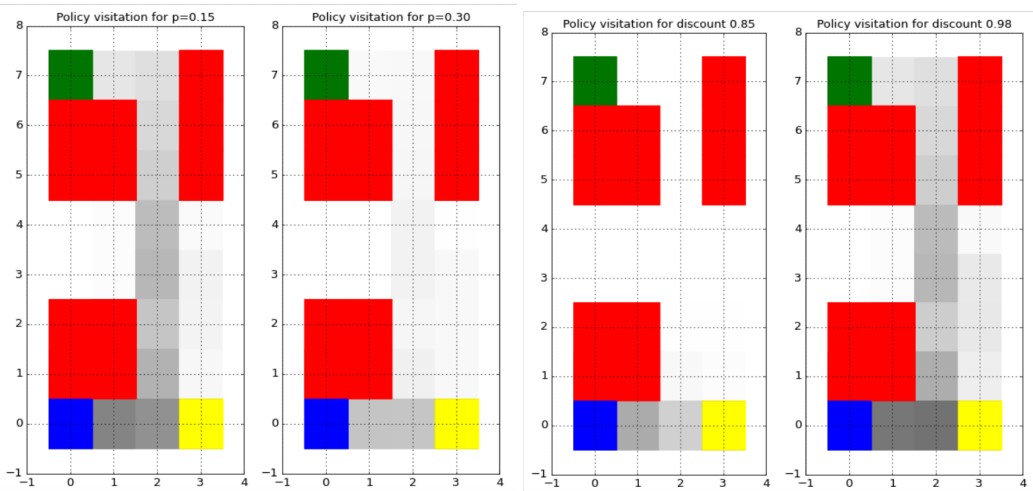

**Figure 10:** Visualization of gridworld policies (as state-visitation distributions) with (a) different transition biases (probability $p$ of unintended transitions), and (b) different discount factors.

## D  EXPERIMENT DETAILS

### D.1  TABULAR EXPERIMENTS

**Environment and training details**  Recall that the gridworld environments we consider are described by $8 \times 4$ grids, with a start and goal state, and walls, lava, and exactly one waypoint state placed in between. We consider a sparse reward where the agent earns a reward of $\theta = 3$ upon reaching the goal state. Alternatively, if the agent reaches a lava or waypoint state, then its reward is 0 or 1, respectively, for the rest of the trajectory. The agent is able to move in either of the four direction (or choose to stay still), and there is a $p = 30\%$ chance that the agent travels in a different direction than commanded. We choose $\gamma^* = 0.98$ high enough that the goal state is preferred over the closer waypoint state under the optimal policy.

A reward-conditioned policy (model or demonstrator) under each environment is given by $\pi(a \mid s; \theta) \propto \exp(Q(s, a; \theta))$, where $Q(s, a; \theta)$ were derived by value iteration using the an MDP model (can be the true underlying MDP or a biased one) of the environment. During reward inference, we discretize the reward parameter space $\Theta = [1, 4]$ with resolution 64. Because the environment is tabular, instead of sampling demonstrations $\mathcal{D}$ from $\pi$, we can instead compute $w^\pi$ the discounted stationary distribution. Specifically, let $\rho$ be the distribution of the starting state (which in our case, is an indicator vector at the start state of each environment), then $w^\pi$ satisfies:

$$w^\pi(s) = (1 - \gamma)\rho(s) + \gamma^* \sum_{s' \in \mathcal{S}} \sum_{a' \in \mathcal{A}} w^\pi(s')\pi(a' \mid s')P(s \mid s', a').$$

We can use this to solve for the true state visitations $w^*$ for any demonstrator policy $\pi^*$, which can be used to compute the weighted policy divergence as in (3) without explicitly sampling a dataset $\mathcal{D}$ of demonstrations.

**Visualization of biased policies**  In Figure 10, we visualize the demonstrator policies $\pi^*$ under the systematic biases considered. We see that in Figure 10(a), when the demonstrator underestimates the probability of unintended transitions, it heavily prefers the goal state, which has higher reward but is much more dangerous to reach, over the waypoint state Conversely, in Figure 10(b), when the demonstrator underestimates the discount factor, they strongly prefer the waypoint state that yields lower reward but is much closer.

### D.2  CONTINUOUS CONTROL EXPERIMENTS

**Environment and training details**  Recall that the domain we consider is the Lunar Lander game, where an agent needs to navigate a lander onto the landing pad. The reward function yields a large

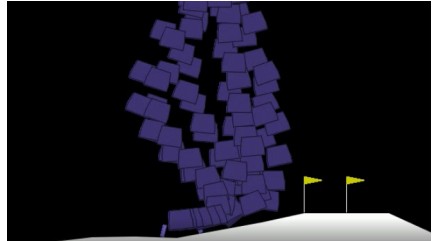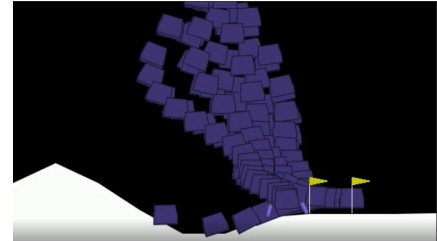

**Figure 11:** Visualization of Lunar Lander trajectories for policies with (a) biased internal dynamics that underestimate left-right acceleration and (b) correct internal dynamics.

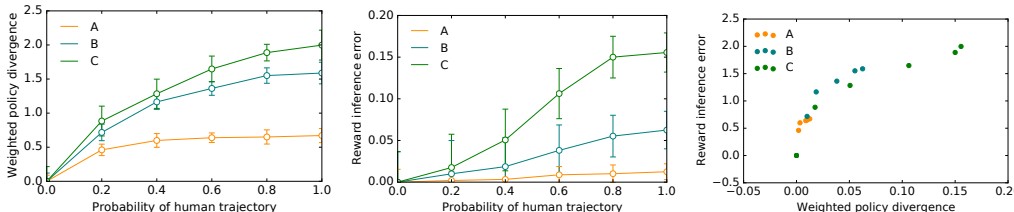

**Figure 12:** Effect of human bias (measured by probability of acting according to human policy) on (a) weighted policy divergence and (b) reward inference error on discrete Lunar Lander environments. In (c), we show a scatter plot of the policy and reward errors for fixed probabilities.

reward for landing on the pad, and a penalty for crashing or going out of bounds. The magnitude of the reward depends on the speed and tilt of the lander upon reaching the landing pad. The physics of the game are deterministic. The reward parameter $\theta \in [0, 1]$ we try to infer is the location of the landing pad (expressed as normalized horizontal displacement).

In this domain, we train a reward-conditioned policy $\pi(a \mid s; \theta)$ by folding the reward parameter $\theta$ into the state representation, which is an $8$-dimensional vector capturing the lander's current location, velocity, and tilt. The policy is parameterized as a 3-layer fully-connected neural network with hidden dimension of $128$, and outputs a squashed Gaussian distribution over actions. Because the state and action space are continuous, we use soft-actor-critic (SAC) (Haarnoja et al., 2018) with fixed entropy regularization $\alpha = 1$. We train the policy for $600$ episodes of length at most $1,000$, with a batch size of $264$, until the policy was able to land on the landing pad with a high success rate. During reward inference, we discretize the reward parameter space $\Theta = [0, 1]$ with resolution $32$. We sample datasets $\mathcal{D}$ consisting of $10,000$ observations, and report the mean and standard error of policy and reward error across $10$ independent samples of datasets.

**Visualization of biased policies** In Figure 11, we visualize the demonstrator policies $\pi^*$ under different degrees of internal dynamics bias. Recall that parameter $p$ describes how much one unit of power will increase acceleration in the left-right directions. When $p$ is underestimated, the policy will not move right enough to reach the landing pad; in contrast, when $p$ is properly estimated, the policy will reach the landing pad with a high success rate.

### D.3 Additional experiment with human policies

In line with the experiments with simulated biases, we run an additional experiment similar to the one in Section 6.3, where we instead keep $\tilde{\pi}$ fixed as the optimal policy, and interpolate between the optimal policy and the real human policy for demonstrator policy $\pi^*$. We show the effect of the interpolation proportion on policy divergence and reward inference error in Figure 12. Again, we notice the consistent message that policy error bounds reward error.

