# OpenReview forum: "On the Sensitivity of Reward Inference to Misspecified Human Models"
_ICLR.cc/2023/Conference — ICLR 2023 notable top 5%_

### Official Review · Reviewer_PHhN · 2022-10-25

**Confidence:** 3
**Correctness:** 4
**Technical Novelty And Significance:** 3
**Empirical Novelty And Significance:** 3
**Recommendation:** 8

**Clarity, Quality, Novelty And Reproducibility:**

Clarity: The paper uses a highly clear presentation of the motivation, method and results. The problem is nicely defined and situated into available work. The theoretical and empirical results are clearly described as well.
Quality: The analysis and derived bound is clearly motivated and formalized. The taken assumptions, possible limitations as well as important next steps are also highlighted.
Novelty: The paper provides a novel analysis on reward functions, focussing on bounding the inference error for close reward functions
Reproducibility: The empirical analyses are well-documented, such that evaluations with biases on the mentioned environments can be reproduced.

**Strength And Weaknesses:**

Pros:
* High clarity wrt motivation, related work, assumptions and conducted theoretical and experimental work
* Important topic - it is a baseline for numerous human-centered RL approaches focussing on AI alignment
* Careful evaluation

Cons:
* Minor open questions to reward comparison / generalization to other reward learning settings. How, for example, does recent work on  equivalent-policy invariant comparison fit compared to the chosen reward similarity? Would it be possible derive similar results here or are such approaches prone to get caught in problems for adversarial samples? Is the derived bound dependent on chosen IRL methods / methods that assume a dataset of expert states and actions or would it transfer to reward functions learned based on trajectory comparisons or other kinds of datasets?
* The related work could also other works on reward learning, e.g. from human preferences

**Summary Of The Paper:**

The paper theoretically and empirically targets the question how/if small errors in learned (human) reward models can lead to large inference errors, thereby making them unusable. The goal then becomes to try to bound the inference error by a function to guarantee usability. The authors argue that it is theorerically provable that the inference error (using maximum likelihood estimation) can be arbitrarily large for sufficiently close human reward functions (learned on finite datasets) for worst-case policy divergence, but that the case is unlikely for "non-adversarial" sampled datasets compared under a weighted policy divergence. The paper identifies central assumptions for the human reward functions, which lead to a possible bound of the inference error. The authors also provide an empirical evaluation for diagnostic grid world domains and LunarLandor for generated as well as real human biases. All experiments support the claims of a bounded inference loss.

**Summary Of The Review:**

The paper clearly describes and situates the targeted problem for learned reward models and inference and provides thorough theoretical and empirical analyses. The problem at hand is important and the contribution has sufficient impact. I am still wondering if/how the results can be applied to other forms of reward learning.

## Post author response
I want to thank the authors for their answers to my questions. Also based the other reviews for the paper, my positive evaluation of the paper remains.

---

> ### Author Response · Authors · 2022-11-12
> **Response to Reviewer PHhN**
>
> Thank you for your review. You raised some very interesting directions for potential future work, which we address below. We also have incorporated discussion of these directions to Section 7 of our paper!
>
> **Generalization to other reward learning settings**
>
> We use a very simple metric for measuring reward learning – squared distance in the reward parameters. This is general, but comes with limitations, such as not being invariant to transformations; we remedy this by adding in Assumption 1, but it is true that better reward distance metrics exist. In particular, EPIC, which you brought up, is one such reward distance that does not share the same limitations. However, it relies on a coverage distribution to weight over the states, which is an environment-dependent quantity. Our choice of reward distance is agnostic to the environment, which is advantageous  to making general arguments on reward learning. We agree that an interesting direction for further analysis could be to consider alternative distances such as EPIC.
>
> **Other works on reward learning**
>
> Our problem formulation –- reward learning from demonstrations – was simply chosen due to its generality. We agree that other forms of reward learning, such as from preferences, rankings, etc., could be very interesting. Namely, since preferences encode more information, it is possible to derive upper-bound results with weaker assumptions than the ones we make. We agree that this is another potentially interesting direction for future work.

---

### Official Review · Reviewer_vrff · 2022-10-25

**Confidence:** 4
**Correctness:** 3
**Technical Novelty And Significance:** 4
**Empirical Novelty And Significance:** 3
**Recommendation:** 8

**Clarity, Quality, Novelty And Reproducibility:**

The writing is clear and the theory well motivated which was helpful. However, log-concavity could be better explained, e.g. with (counter)examples in the main text. (For example, I wonder if log-concavity is always satisfied by optimal policies?) Furthermore, it would be useful to summarize the intuitions for Theorem 2 and 3 without reference to the appendix.

In the literature on reward inference, this appears to be the first paper to study the relationship between errors in the human model and inferred reward quantitatively. (Though some similar work has seemingly been done in inverse problems). However, the relationship to arXiv:1712.05812 should be discussed. That work implies that inferring the human model (without knowing the reward function) is difficult, meaning that large errors are possible.

Minor issues:

The abstract and introduction should probably be shortened for easier readability.

The supplement contains an old and unfinished version of the paper which was confusing; please check the submission carefully.

The terminology of "human model" is overloaded since it can also refer to the human policy (mapping states to actions rather than reward functions to actions). Other possibilities include human decision model, decision model, decision rule, or decision algorithm.

If one exists, it would be helpful to discuss any known human bias which breaks the assumptions of Theorem 3.


**Strength And Weaknesses:**

The paper studies an important problem: To infer reward functions, a human model is necessary, but it is infeasible to infer it accurately. This is because inferring it would at least require the human's reward function, which is initially unknown. This problem becomes more important as AI systems attempt more complex tasks where human behavior is nowhere near optimal and cannot be modeled as optimal.  If we know the relationship between reward errors and human model errors, we can better understand if we can improve reward learning by learning better human models. This understanding also helps clarify if reward learning is a viable strategy at all for complex tasks where humans are increasingly suboptimal. Both the optimistic and pessimistic result yield useful insight on these questions. The results are clear and succinct and to my knowledge novel, providing a clearly significant improvement in our knowledge of the important question at hand.

Weaknesses

In the first and last section, the authors make an optimistic conclusion overall, saying that their results imply that reward inference with misspecified human models is not very difficult. The positive conclusion needs some strong caveats stated in the first and last section to avoid false impressions. Though I think these caveats should be stated, I don't think they detract, and in fact add to, the value of the paper. By order of importance:

Firstly, the introduction speaks of mild assumptions for the positive conclusion that a slightly misspecified human model will lead to small errors in the inferred reward. However there is an assumption baked into the choice of error metric for human models: the KL(human_policy | model_policy). To assume that the human model error is small, this KL divergence assumes that the human policy would *never take an action that the human model places ~0 probability on*. Otherwise, the KL will be large, allowing large reward errors, and challenging the paper's positive conclusion.

Aside from noting this caveat, the authors could reproduce their results with a sufficiently different metric such as the reverse KL. Furthermore, the authors could note that having a small forward KL requires that the human model 'covers' the support of true human, as otherwise the KL would be large. This is an interesting result, as it means that, all we need is to learn human models with good coverage. (And it is not a problem if the human model sometimes takes actions that humans would not take).

Second, the authors construct a case where the inference error is large, but then argue that their construction is unlikely to occur in practice and that this supports their optimistic conclusion. However, they have not ruled out that other constructions exist. Some of these may be more likely in practice.

Third, the paper could justify its choice of reward distance metric, the squared distance between reward parameters. In particular, a small error in inferring the reward parameters might still lead to a large error in the resulting return and policy.




**Summary Of The Paper:**

Inferring human reward functions requires 'human models' which specify how a human acts given their reward function. This paper studies how errors in the human model translate into errors in the inferred reward. First it shows a pessimistic result: in an adversarial setting, a small error in the human model can lead to a large error in the inferred reward. However, under some assumptions the paper shows that the reward error can be bounded linearly in the human model error (for particular choices of error metrics). The paper show that this result applies to several known human biases. The result matches up with experiments in two environments. The authors thus make an optimistic conclusion: reward inference with misspecified human models is not very difficult.

**Summary Of The Review:**

The paper makes a significant contribution to filling an important knowledge gap: can we accurately infer reward functions with a slightly misspecified human model? The theoretical results are clear and useful. While the overall positive conclusion seems to need caveats, these do not affect the strength of the contributions. If the caveats can be clarified, I would consider raising my score since I believe the paper will be among the more useful ones accepted to ICLR.

EDIT: The authors partially addressed the suggestions I made and argued for the generality of the results (for example by noting that the KL upper bounds the TV distance). This is one of the somewhat rare papers in modern ML where a clean theoretical result provides a clear and non-obvious practical benefit. In addition, the benefit pertains to a problem that is becoming increasingly important. Assuming the authors make the minor changes I suggested, I think the paper should be considered for an award.

---

> ### Author Response · Authors · 2022-11-12
> **Response to Reviewer vrff**
>
> Thank you for your detailed review! We agree with many of your insights, specifically on improving the discussion of the modeling assumptions that we make, and have incorporated them into our paper (more details below).
>
> **KL DIVERGENCE AS A DISTANCE METRIC**
>
> You are correct that our choice of KL-divergence (and its direction) comes with implications, specifically that the human model must cover the actions taken by the human. This is insightful, and we have added a discussion of this to Section 3 of our paper. We specifically choose the forward direction of KL-divergence because it contains an expectation over the true human policy. This aligns with the dataset, which is collected from demonstrations of the true human!. We agree that similar results can be derived using reverse KL-divergence, where the expectation over the true human would need to be replaced by the human model. We simply find this unnatural because the dataset is derived from the true human.
>
> **ADVERSARIAL EXAMPLE LEADING TO LARGE REWARD ERROR**
>
> You are of course absolutely correct in that other examples outside of the one that we present may exist that lead to large reward inference error. We did not aim for the unlikeliness of this example to act as evidence of our positive result. We simply wanted to point out that such examples exist, but this particular one was constructed adversarially. We will rewrite our conclusions in Section 4, where the adversarial example was introduced, to make this more clear. And we will re-emphasize that our positive result does require a strong assumption on log concavity, so we have no guarantees outside of that, given the negative result. We only have the empirical data to send the reassuring message that even when log-concavity does not seem to hold, in practice we do not see small errors in the model leading to huge errors in the reward.
>
> **REWARD PARAMETERS AS A DISTANCE METRIC**
>
> We agree that performance of the learned policy may be a more practical metric to determine the effectiveness of IRL algorithms. However, we believe that one of the central advantages of IRL (that explicitly learns the reward parameters) over imitation learning is that reward parameters can be generalized to new, test environments, i.e. ones with different initial state distributions or transition dynamics. Hence, in our work, we explicitly consider reward parameters in our analysis. You are correct in that similar reward parameters can lead to different policy performance, particularly without some smoothness assumption of the rewards on reward parameters. This is definitely a limitation of looking at the reward parameters. We added a discussion of this in Section 3 of our paper.

---

> > ### Comment · Reviewer_vrff · 2022-11-12
> > **Thanks + suggestions**
> >
> > Thank you for making the clarifications here and in the paper. I think this is an improvement. Since many/most readers only read the abstract and introduction, I think a bit of clarification there would be appropriate to avoid giving a false impression. Some suggestions:
> >
> > - The introduction says "bound the reward inference error by some
> > function of the distance ... for some reasonable definition of distance". In the current version it is unclear if your result holds for one definition of distance or for many. On the first read I falsely though you meant many. You could specify that you mean a specific one, the KL of the human and the human model.
> >
> > - Replacing "reward inference error" with "error in inferred reward parameters" in intro and abstract.

---

### Official Review · Reviewer_Se9W · 2022-10-26

**Confidence:** 2
**Correctness:** 3
**Technical Novelty And Significance:** 3
**Empirical Novelty And Significance:** 3
**Recommendation:** 3

**Clarity, Quality, Novelty And Reproducibility:**

Clarity: The paper is very well written
Novelty: There is no new methodology to evaluate novelty for, so it is mainly the question (how harmful misspecified human models are for reward inference) that is novel
Reproducibility:  Experiment details are included in the appendix.

**Strength And Weaknesses:**


Strengths: the paper is very well-written, and I appreciate the attempt to explore different types of human irrationality (e.g. myopia), and the experiments include real human data (albeit w/ a very simplified form of misspecification via simple re-weighting)

- I am a bit confused why measuring reward misspecification by distance in parameter space works except for very restrictive, simple linear reward models. If the reward function is a NN, then it is well known that two networks that behavior similarly can have widely different parameters. Any necessary assumptions here should be clearly stated.
- I am likewise confused why KL diverge between demonstration policy and the human model is the right way to measure distance. Given the motivations discussed in the paper of modeling humans via noisy-rationality, systemic biases, etc., it seems like humans may deviate from an idealized model in ways much stronger and more state-specific than the KL measure assumes.
While I like the idea of fitting certain types of irrationalities (myopia and illusion of control) into this framework, it feels a bit cherry-picked to only study irrationalities that satisfy the specific notion of distance the authors chose.

**Summary Of The Paper:**

The paper addresses the question of whether misspecification of human behavior models for IRL can lead to detrimental effects on reward inference accuracy, and provide theoretical guarantees showing that while adversarial situations can be constructed such that small human behavior models lead to "catastrophic" inference errors, under a log-concavity assumption the effect is more stable. The authors then provide empirical results showing a correlation between human model accuracy and reward inference accuracy.

**Summary Of The Review:**


My primary concern and reason for rejection is the rather simplified assumptions the authors make in order to "get the theory to work".  It is also unclear what the ultimate takeaway here is - if the goal of studying the sensitivity of reward inference is to make some kind of claim on whether reward learning is a valid direction, then only studying restricted forms of human model misspecification does not really lead to any meaningful insights. I think a more clear, isolated section stating the author's assumptions is necessary for acceptance.

---

> ### Author Response · Authors · 2022-11-12
> **Response to Reviewer Se9W (1/2)**
>
> Thank you for your review! We take this opportunity to make a clarification that might somewhat change your perspective on the work.
>
> **GENERALIZABILITY OF THE RESULT TO DIFFERENT MISSPECIFICATION**
>
> It looks like a primary concern goes back to the KL divergence as an error metric on the human model: “if the goal of studying the sensitivity of reward inference is to make some kind of claim on whether reward learning is a valid direction, then only studying restricted forms of human model misspecification does not really lead to any meaningful insights”.
> Our paper arguably does the opposite: we study any kind of misspecification, and instantiate our general result for a few plausible kinds.
>
> While we can enumerate certain biases based on behavioral econ findings, the reality is that we have no idea how humans actually behave, i.e. our models will be misspecified in all likelihood in ways that are very different from, e.g., assuming noisy-rationality when the human is myopic. Therefore, we contribute a bound based on a very general notion of error, that is not bias-specific.
>
> KL divergence is arguably natural because the policies as probability distributions. We chose KL divergence over other measures (i.e. TV, Hellinger, etc.) simply because it was a natural metric and often used in prior works as objective to minimize distance between distributions, e.g. in variational inference. Note that KL divergence can be used to upper-bound other divergence metrics such as TV via Pinkster’s inequality, so we do not view our specific choice of divergence as limiting. The conclusion is that improving this KL divergence should improve the worst-case reward error too. This is what we should base any conclusion about the validity of reward learning, rather than the few specific biases we later analyze.
>
> We then specialize this general result to specific biases as examples to provide intuition on how these divergence measures scale, i.e. how they relate to intuitive parameters we can understand (delta in the transition model, delta in the horizon). But we do not think it is fair to say that our paper only applies to a few biases, since the main result is a statement on how the reward error is bounded by a natural measure of model error.
>
> **REAL HUMAN DATA RESULT VIA “SIMPLE REWEIGHTING”**
>
> We empirically test how the reward error behaves as we go further away from the true human policy (in a specific environment). It is true that we interpolate between the human policy and the noisy-rational assumption. However, it is important to clarify that as we get closer and closer to noisy-rationality, this captures really complex and nuanced forms of misspecification, as the real human policy we collect is whatever real people do, and is likely the result of a number of unidentified biases. This misspecification is as realistic as it gets! As for the degree of misspecification being dictated by the weighting, we agree this is a simple way to do it, but it emulates the process of fitting a better and better model (e.g. imagine getting more and more human data).
>
> **REWARD PARAMETERS AS A DISTANCE METRIC**
>
> We wholeheartedly agree with you that there are limitations to evaluating distance between the reward parameters. We choose reward parameters over return of the learned policy because it is agnostic to the test environment. Namely, reward parameters can be used to generalize to different test environments i.e. ones with different initial state distributions or transition dynamics. We view this as one of the major advantages to explicitly learning reward parameters, over imitation learning that would directly learn a policy.
>
> However, it is true that different reward parameters can lead to drastically different reward functions. We remedy this via Assumption 1, which states that unique reward parameters exist that explain the behavior in the data. We could relax Assumption 1 with a refined analysis that would consider equivalent classes of reward parameters; however, we believe such analysis is orthogonal to conveying the message of our work, that learning reward parameters can be promising even under misspecification.
>
> Regarding your comment on a “clear, isolated section,” for assumptions, we try to accomplish this via Section 3, which contains the modeling assumptions that we make. The only one that is not in Section 3 is Assumption 2 in Section 5, because it is only used to derive our upper-bound.

---

> > ### Author Response · Authors · 2022-11-12
> > **Response to Reviewer Se9W (2/2)**
> >
> > **SUMMARY**
> >
> > Our work does not apply to only specific biases, rather derives a general upper bound based on divergence. We show that this relates to a few of the very large set of biases identified by behavioral econ, in order to build intuition on what the metric does. We test it empirically with data generated from these biases, but, more importantly, also with real human data – showing that our upper bound is somewhat predictive to what happens as you introduce whatever unidentified, nuanced, complex biases are present in that real human data. On the other hand, we definitely agree that parameter error is not ideal, but neither is policy return in a known set of environments (because in reality we never know the test environments).

---

> ### Comment · Reviewer_vrff · 2022-11-12
> **Bounding distance in parameter space more useful than it seems**
>
> > I am a bit confused why measuring reward misspecification by distance in parameter space works except for very restrictive, simple linear reward models. If the reward function is a NN, then it is well known that two networks that behavior similarly can have widely different parameters. Any necessary assumptions here should be clearly stated.
>
> Firstly I'll assume you meant "two networks with similar parameters can have widely different behavior" which means that bounding the distance in parameter space may not be very reassuring.
>
> I noted the same criticism in my review and I think it's a reason why the authors could tone down their positive result a bit in the intro/abstract.
>
> But their result is actually stronger than I thought at first, because it applies to EVERY parameterization of the reward function. A given fixed reward function R(s,a) can be parameterized in many ways, e.g. we can use the parameters of a NN but we can also parameterize it by the _graph_ of the function such that the set {r(s,a): s,a in S, A} is the set of parameters. This way, a small change in a parameter cannot lead to a big change in reward, because the parameter _is_ the reward. Further, a given NN can be parameterized in many ways, for example Weight Normalization. An interesting reparameterization is to use parameters theta' = theta*V where V is a diagonal square matrix that stretches every parameter (to avoid changing the function, theta' is multiplied by V^-1 in the NN architecture). This way, we can address the example where a small change in one parameter theta[i] leads to a big change in reward: we just stretch theta[i] to be theta'[i]=theta[i]*v[i] for some large v[i]. Now a small change in theta[i] is a large change in theta'[i], meaning that theorem 3 forbids it. So bounding distance in parameter space means that all parameterizations of the reward function have small reward inference error, which is more reassuring.

---

> ### Author Response · Authors · 2022-11-17
> **Let us know if you have further concerns.**
>
> In the remaining time, we would appreciate it if you let us know if you have any remaining questions/concerns, as we would be happy to address them!

---

> ### Author Response · Authors · 2022-12-10
> **Let us know if you need anything else.**
>
> Thanks for reviewing our work. Before discussion ends, we would appreciate it if you let us know what your remaining concerns are with our work. We will do our best to answer them!

---

### Official Review · Reviewer_tbzC · 2022-10-29

**Confidence:** 3
**Correctness:** 4
**Technical Novelty And Significance:** 3
**Empirical Novelty And Significance:** 3
**Recommendation:** 8

**Clarity, Quality, Novelty And Reproducibility:**

The paper is clear in most parts (except the details of the human experiment). I also like both theoretical and experimental contributions. Given the code, simulations are reproducible, but I did not find any statement on the availability of human data.

**Details Of Ethics Concerns:**

I did not find information on human data (maybe I missed it).
UPDATE: My previous concerns around Ethics/IRB has been resolved.
To authors: In the de-anonymized version, please state the institute that issued the approval.

**Strength And Weaknesses:**

Strength:
The paper tackles an important problem. It contains both theoretical and empirical results. Moreover, the experiments contain both simulation and real data.

Weakness:
 My main concern: I did not find IRB approval information on the human experiment. If there is, it should be mentioned, if not the authors should explain why it is not necessary in this case (and should be validated with the conference chairs). Also, the details of the experiment and instructions to the demonstrators should accompany the paper.

Worst case v.s. average (end of page 3, beginning of page 4): If I understood correctly the worst-case error is based on all states but the average is based on available states in the data. This looks inconsistent, what is the rationale behind it?


**Summary Of The Paper:**

This work analyzes the effect of model accuracy on reward inference accuracy when fitted to human behavior. Specifically, it shows it is possible that a small model error leads to a very large error in inferring the reward. However, this scenario is unlikely. The authors backed their claims with simulations and real human data.

**Summary Of The Review:**

It is a good solid paper to me both in terms of theoretical analysis and experimental results, but human data experiment should be more clear.

---

> ### Author Response · Authors · 2022-11-12
> **Response to Reviewer tbzC**
>
> Thank you for your review. You have raised some important concerns that we aim to address below.
>
> **HUMAN EXPERIMENTS**
>
> Yes , we do have an IRB protocol approved by our institution. We can include any details the reviewer thinks are appropriate in the camera-ready (or simply clarify that we collected participant data under an IRB-approved protocol).
>
> **WORST CASE VS. AVERAGE**
>
> You are correct in pointing out the difference between the two policy divergence metrics that we use. Note that because the worst-case divergence is max over states, whereas weighted divergence is an average over them, we have that weighted divergence <= worst-case divergence.
>
> We use two different divergences to strengthen our results.  Namely, our lower-bound shows that worst-case divergence can be small, but reward inference error is still large. This is strictly a stronger claim that if we were to use weighted divergence because we use the larger of the two divergences. Similarly, our upper-bound shows that under some assumptions, reward inference error is linearly bounded by weighted divergence. There, we use the smaller of the two divergences.

---

### Decision · Program_Chairs · 2023-01-20

**Decision:**

Accept: notable-top-5%

**Justification For Why Not Higher Score:**

N/A

**Justification For Why Not Lower Score:**

- this appears to be the first paper to study the relationship between errors in the human model and inferred reward quantitatively

- This is one of the somewhat rare papers in modern ML where a clean theoretical result provides a clear and non-obvious practical benefit. In addition, the benefit pertains to a problem that is becoming increasingly important. I think the paper should be considered for an award.

**Metareview: Summary, Strengths And Weaknesses:**

Summary:

Inferring human reward functions requires 'human models' which specify how a human acts given their reward function. This paper studies how errors in the human model translate into errors in the inferred reward. First it shows a pessimistic result: in an adversarial setting, a small error in the human model can lead to a large error in the inferred reward. However, under some assumptions the paper shows that the reward error can be bounded linearly in the human model error (for particular choices of error metrics). The paper show that this result applies to several known human biases. The result matches up with experiments in two environments. The authors thus make an optimistic conclusion: reward inference with misspecified human models is not very difficult.

Strengths:

- The writing is clear and the theory well motivated.
- this appears to be the first paper to study the relationship between errors in the human model and inferred reward quantitatively
- This is one of the somewhat rare papers in modern ML where a clean theoretical result provides a clear and non-obvious practical benefit. In addition, the benefit pertains to a problem that is becoming increasingly important. Assuming the authors make the minor changes I suggested, I think the paper should be considered for an award.
- High clarity wrt motivation, related work, assumptions and conducted theoretical and experimental work
- Important topic - it is a baseline for numerous human-centered RL approaches focussing on AI alignment
- Careful evaluation

Weaknesses:

- In the first and last section, the authors make an optimistic conclusion overall, saying that their results imply that reward inference with misspecified human models is not very difficult. The positive conclusion needs some strong caveats stated in the first and last section to avoid false impressions. Though I think these caveats should be stated.

- log-concavity could be better explained,

Recommendation:

A majority of reviewers vote strongly for acceptance. I, therefore, decide to accept the paper. I encourage the authors to use the feedback provided to improve the paper for its camera ready version.


**Note From Pc:**

if the above contains the word "oral" or "spotlight" please see: "oral" presentation means -> notable-top-5% and "spotlight" means -> notable-top-25%. As stated in our emails, we are disassociating presentation type from AC recommendations